# Methicillin-Resistant *Staphylococcus aureus* from Diabetic Foot Infections in a Tunisian Hospital with the First Detection of MSSA CC398-t571

**DOI:** 10.3390/antibiotics11121755

**Published:** 2022-12-04

**Authors:** Ameni Arfaoui, Rym Ben Sallem, Rosa Fernández-Fernández, Paula Eguizábal, Raoudha Dziri, Idris Nasir Abdullahi, Noureddine Sayem, Salma Ben Khelifa Melki, Hadda-Imen Ouzari, Carmen Torres, Naouel Klibi

**Affiliations:** 1Laboratory of Microorganisms and Active Biomolecules, Faculty of Sciences of Tunis, University of Tunis El Manar, Tunis 1068, Tunisia; 2Biochemistry and Molecular Biology, University of La Rioja, 26006 Logroño, Spain; 3Service of Biology, Carthagene International Hospital of Tunisia, Tunis 1082, Tunisia

**Keywords:** diabetic foot infection, *S. aureus*, MRSA, MSSA-CC398

## Abstract

This study sought to analyze the antimicrobial resistant phenotypes and genotypes as well as the virulence content of *S. aureus* isolates recovered from patients with diabetic foot infections (DFIs) in a Tunisian hospital. Eighty-three clinical samples of 64 patients were analyzed, and bacterial isolates were identified by MALDI-TOF. The antimicrobial resistance phenotypes were determined by the Kirby–Bauer disk diffusion susceptibility test. Resistance and virulence genes, *agr* profile, *spa* and SCC*mec* types were determined by PCR and sequencing. *S. aureus* was detected in 14 of the 64 patients (21.9%), and 15 *S. aureus* isolates were recovered. Six out of the fifteen *S. aureus* isolates were methicillin-resistant (MRSA, *mecA*-positive) (40%). The isolates harbored the following resistance genes (number of isolates): *blaZ* (12), *erm*(B) (2), *erm*(A) (1), *msrA* (2), *tet*(M) (2), *tet*(K) (3), *tet*(L) (1), *aac*(6′)*-aph*(2″) (2), *ant*(4″) (1) and *fex*A (1). The *lukS/F-PV* and *tst* genes were detected in three isolates. Twelve different *spa*-types were identified and assigned to seven clonal complexes with the predominance of *agr*-type III. Furthermore, the SCC*mec* types III, IV and V were found among the MRSA isolates. Moreover, one MSSA CC398-t571-*agr*-III isolate was found; it was susceptible to all antimicrobial agents and lacked *luk-S/F-PV, tst, eta* and *etb* genes. This is the first report on the prevalence and molecular characterization of *S. aureus* from DFIs and also the first detection of the MSSA-CC398-t571 clone in human infections in Tunisia. Our findings indicated a high prevalence *S. aureus* in DFIs with genetic diversity among the MSSA and MRSA isolates.

## 1. Introduction

*Staphylococcus aureus* (*S. aureus*) strains have become a leading cause of hospital-associated and community-associated infections worldwide. Two mechanisms confer resistance to β-lactams in staphylococci with the most common being the production of β-lactamase, encoded by *bla*Z, which produces the hydrolysis of the β-lactam ring, rendering the β-lactam inactive. More than 90% of staphylococcal isolates now produce penicillinases [1]. The second mechanism is due to an altered penicillin-binding protein PBP2a, encoded by *mec*A, which is carried in a variable mobile element, namely, the staphylococcal chromosome cassette *mec* (SCC*mec*) [2]. This mechanism leads to resistance to a semi-synthetic penicillinase-resistant β-lactam called methicillin. Furthermore, the term methicillin resistance manifests as resistance to virtually all β-lactams with the exception of the latest generation of cephalosporin β-lactams [2]. MRSA strains can also acquire additional resistance to several commonly used non-β-lactam antimicrobials (e.g., aminoglycosides, macrolides, fluoroquinolones and tetracycline) and are currently considered as the first class of multidrug-resistant (MDR) pathogens [3].

Diabetic foot infection (DFI), defined as soft tissue or bone infection below the malleoli, is the most common diabetic complication that often leads to hospitalization and non-traumatic lower extremity amputation [4,5,6]. Many studies have shown that DFI is polymicrobial [7,8,9]. Particularly, *S. aureus* is the bacteria implicated the most [10]. In addition to its ability to acquire antimicrobial resistance (AMR) to many clinically important drugs [11], this microbe plays a significant role in DFIs by causing infections ranging from superficial to severe and potentially fatal systemic infections [12]. 

The overuse of antibiotics is one of the most serious issues with DFI treatment. The prescription of unsuitable antimicrobial treatment has an impact on the microbiota and encourages the selection and growth of MDR bacteria. Thus, the global emergence of MRSA has considerably restricted the available therapeutic options for staphylococcal infections [13].

The epidemiology of AMR in Tunisia has been very dynamic in recent years, and the available data give insights into the alarming situation, especially in hospital settings [14]. In contrast, only three earlier studies have been conducted on DFI. These studies were limited to the bacteriological profile of DFI patients and showed controversial results. *Enterobacteriaceae* were the main bacteria causing the infection in diabetics in two of them [8,9], while *S. aureus* was the most frequent pathogen isolated in the remaining one [15]. Thus, there are currently no data on the molecular characterization of the bacterial strains involved, risk factors or treatment of multi-drug resistance organisms in patients with DFI in Tunisia. The aim of the present study was to evaluate the prevalence of *S*. *aureus* isolated from diabetic patients admitted for infected foot ulcers in the multidisciplinary diabetic foot center of the International Hospital, Carthagene in Tunisia during the COVID-19 pandemic and to investigate their genetic relatedness, antibiotic resistance pattern and virulence characteristics.

## 2. Results

### 2.1. Patient Characteristics and MRSA Prevalence in Ulcer Samples

As shown in Table 1, 64 patients were included in this study (48 men and 16 women with a mean age of 62.28 years); 6 of them had diabetes type 1 (4 male and 2 female), 57 diabetes type 2 (43 male and 14 female) and 1 male had diabetes secondary to acute pancreatitis. Since the diabetic foot center is international, the patients were from different African countries (Tunisia, Algeria, Libya, Chad and Guinea). Accordingly, the distribution of the nationality among the patients with foot ulcer infection was as follows: Tunisia (51.6%), Libya (39%), Algeria (6.3%), Chad (1.6%) and Guinea (1.6%).

*S. aureus* was detected in 21.9% of all patients with DFI analyzed in this study (14/64). One isolate per positive sample was included, except in one patient in which two different isolates were recovered and both of them were included making a collection of fifteen *S. aureus* isolates. Six of these fifteen *S. aureus* isolates were MRSA (cefoxitin-resistant and *mec*A positive) (40%), and the remaining nine isolates were methicillin-susceptible *S. aureus* (MSSA). One of the patients carried one MSSA and one MRSA isolate. Thus, the overall prevalence of MRSA in the ulcers was 9.4% and reflected a proportion of 42.9% of the participants who had *S. aureus* foot ulcer infections.

### 2.2. Antimicrobial Resistance Pattern of the S. aureus Isolates

The resistance profiles of MRSA and MSSA isolates to the antimicrobial agents tested are presented in Table 2. All fifteen isolates showed resistance to at least one antibiotic. Multidrug resistance was found in 53.4% of isolates. All six MRSA isolates showed resistance to fusidic acid, four isolates to tetracycline and three isolates to tobramycin and gentamicin. Two MRSA isolates were resistant to ciprofloxacin, levofloxacin, trimethoprim–sulfamethoxazole and erythromycin, and one to clindamycin, minocycline, mupirocin and rifampicin. All the MRSA isolates were susceptible to tigecycline, vancomycin, teicoplanin and linezolid.

Among the nine MSSA isolates, antimicrobial resistance was only shown against penicillin (88.9%), fusidic acid (55.6%) and erythromycin (22.2%).

### 2.3. Genotypic Patterns of Antibiotic Resistance among S. aureus Strains

The *mec*A gene was found in the six MRSA isolates (40%), and five of these isolates also carried the *blaZ* gene (encoding penicillin resistance). All tetracycline-resistant isolates carried *tet* genes (*tet*(M) (n = 1), *tet*(K) (n = 1), *tet*(L) and *tet*(K) (n = 1) and *tet*(M) + *tet*(K) (n = 1)). Concerning the three tobramycin- and gentamicin-resistant isolates, two harbored the *aac*(6′)-*aph*(2”) gene, and one co-harbored the *aac*(6′)-*aph*(2″) and *ant*(4′)-Ia genes.

Among the nine MSSA isolates, the *blaZ* gene was found in seven isolates (77.8%). The presence of *erm*(B) alone or in association with *erm*(A) was detected in two erythromycin-resistant isolates. The *msr*A gene was identified in two isolates and the *fexA* gene in one isolate. Two strains had no resistance genes (Table 3).

### 2.4. Molecular Typing of Isolates

Twelve different *spa* types were identified among the fifteen *S. aureus* isolates. The *spa* type t127 was detected in three isolates, while others were detected only once: (t311, t037, t15077, t688, t084, t188, t355, t091, t012, t223 and t571). One of the fifteen isolates could not match any known *spa* sequence. For one patient, two *S. aureus* isolates were identified corresponding to two different *spa* types (t311 and t571). The isolate belonging to *spa*-type t571 was assigned to the clonal complex CC398.

Among the six isolates that carried the *mecA* gene, two of them harbored SCC*mec* type V, one isolate SCC*mec* type IVb and another SCC*mec* type III, and the remaining two isolates were not typable. The characterization of the *agr* system showed a predominance of *agr* group III (11 isolates, 73.3%). The *agr* group IV, II and I were detected in four isolates.

### 2.5. Virulence Profile

The *lukF/lukS-PV* genes encoding for Panton–Valentine leukocidin (PVL) were detected in two isolates (13.33%) typed as t127-MRSA and t355-MSSA. The *tst* and *eta* genes were found, each in one isolate. None of the strains carried genes encoding the ETB toxin.

In addition, all MRSA and seven MSSA isolates carried the *scn* gene of the IEC system, and they were ascribed to different IEC types (A, B, C, D, E and G). In addition, two MSSA isolates lacked the *scn* gene (IEC-negative). Thirteen isolates (87%) contained an IEC-converting βC-Φs, as demonstrated by the presence of *scn*. The predominant IEC variant was type D (*sea, sak* and *scn*) found in four isolates (30.7%). Variant E (*sak* and *scn*), G (*sep*, *sak* and *scn*), C (*chp* and *scn*), A (*sea, sak, chp* and *scn*) and B (*sak, chp,* and *scn*) were present in three, two, two, one and one isolates, respectively (Table 3).

## 3. Discussion

Methicillin-resistant *S. aureus* is a dominant hospital pathogen in Tunisia and worldwide [16]. Although antibiotic resistance of healthcare-associated staphylococci is well documented in Tunisia, no detailed information is available on antibiotic resistance and the molecular characterization of *S. aureus* isolated from diabetic ulcers. The current study fills this knowledge gap by analyzing the prevalence and molecular characteristics of *S. aureus* isolates in the International Tunisian Hospital, Carthagene. Of note, the patients involved were from different countries (Tunisia, Algeria, Libya, Chad and Guinea), thus reflecting the characteristics of *S. aureus* associated with DFIs not only in Tunisia but also on a wider geographical scale.

Studies have shown that prior use of antibacterial agents, hospitalization, MRSA nasal carriage and chronic wounds are risk factors for MRSA acquisition in patients with DFIs [17].

In our study, the prevalence of *S. aureus* detected in DFIs was 21.9%. Three previous retrospective studies from different Tunisian hospitals showed that *S. aureus* was isolated in 9%, 17% and 31% of DFIs [8,9,15]. This variation in percentages might be due to a difference in the geographical areas, the method applied to obtain cultural samples and study periods, especially the case of our study which took place during the COVID-19 pandemic and partly explaining the decrease in the number of consulting patients.

The prevalence of MRSA in this study was six out of fifteen *S. aureus* isolates (40%), which is a cause for concern given the high clinical significance of this pathogen. Similar studies have been performed in other countries with different prevalences of *S. aureus* in DFIs / prevalence of MRSA among *S. aureus*: Ghana (19%/6%), Iran (46.10%/19.48%), Iraq (38.7%/45.8%), Morocco (12.6%/4.7%), Turkey (20%/31%), Jordan (14.2%/93%) and Algeria (30.7%/85.9%) [18,19,20,21,22,23,24]. A recent meta-analysis including 112 studies from a wide range of countries reported *S. aureus* isolate detection in 109 studies representing 15,670 clinical samples; the proportion of MRSA among these isolates was 18.0% [25].

In our study, multi-resistance was found in 53.4% of isolates. The MDR phenotype is considered a common trait of MRSA isolates from various origins with resistance to several clinically relevant antimicrobial agents typically due to the acquisition of various mobile genetic elements (plasmids and transposons) causing treatment failure and significant associated human health burdens and healthcare costs [26,27]. In Tunisia, the high proportion of isolates showing this resistance phenotype may be related to the abuse of antibiotics with a frequent practice of self-medication. In this study, vancomycin, tigecycline, teicoplanin and linezolid were effective against all the *S. aureus* isolates. Fortunately, these antibiotics remain the best option to treat MRSA-associated infections, thus appropriate use of these antibiotics is highly recommended to avoid the selection of resistant strains.

Overall, high genetic diversity was found among the *S. aureus* isolates demonstrated by thirteen *spa* types and four *agr* groups with the predominance of *agr-*type III and *spa*-type t127 (CC1). These results are consistent with those of a recent review which highlighted that *S. aureus* strains isolated from diabetic foot ulcers in different countries are genetically diverse [28].

The *spa* type t127 was the predominant (three isolates, 20%). It was previously reported that t127 is associated with serious human infections in the United States and Germany [29,30,31]. In addition to a clinical origin, the *spa* type t127 has recently been reported in processed foods in China and in animals, indicating the risk of MRSA transfer from food and animal origins to humans or vice versa [32,33].

Interestingly, an MSSA isolate belonging to the *spa* type t571 (CC398 lineage) was found in one of the patients in our study. To our knowledge, this is the first report of MSSA-CC398 in human infections in Tunisia. Even though livestock-associated (LA)-MRSA-CC398 is closely related to food-producing animals [34], this MSSA-CC398-t571 subclade seems to be livestock-independent and has been detected in human invasive infections in different countries [35,36,37]. Similarly, a national French study showed a consistent and significant association between MSSA-CC398 and diabetic foot osteomyelitis [38].

The Panton–Valentine leukocidin is the most studied toxin produced by *S. aureus* [39]. In our study, only two *S. aureus* isolates contained the genes of PVL (13.3%). It has been suggested that PVL-positive strains are less frequently detected among DFIs as this gene is mostly prevalent in community species [20]. However, some studies reported higher PVL gene rates reaching 14.1% and 57% in Algeria and India, respectively [24,40].

The *tst* gene, encoding toxic shock syndrome toxin-1, was found in one isolate. Other studies reported different rates of *tst* positive strains in cases of diabetic foot ulcers ranging from 13% to 19% [40,41,42], whereas no strains of *S. aureus* from DFIs were positive to *tst* in a previous African study [43].

Among our fifteen *S. aureus* isolates, one of them (6.7%) harbored the *eta* gene and none harbored the *etb* gene. This result was similar to a previous report from France showing that 3% of *S. aureus* strains from diabetic foot ulcers were *eta*-positive, but no strain harbored the *etb* gene [41]. Another European study noted that 13% and 17% of the strains from DFIs harbored *eta* and *etb* genes, respectively [42]. However, these virulence factors were not detected in cases of DFIs previously reported in Algeria [24] or in diabetic foot osteomyelitis in France [38]. *S. aureus* is an extremely versatile pathogen in humans with different virulence phenotypes, suggesting that the virulence determinants did not spread homogeneously among various genetic backgrounds.

## 4. Materials and Methods

### 4.1. Bacteria Collection and Identification

Between September 2019 and October 2020, a total of 83 samples (tissue biopsy and/or deep swab and/or aspiration) were analyzed from 64 patients who were admitted for DFI at the diabetology department of the International Hospital Carthagene of Tunisia; this hospital has a capacity of 300 beds and 55 intensive care beds. Inclusion criteria were diabetic patients with any type of diabetes and aged ≥18. Patients who received antibiotic therapy within 3 months before the consultation, pregnant patients and those with a mental disorder that precluded understanding the scope of the study were excluded from the work.

Samples (aspirations (n = 5), deep swabs (n = 42), and tissue biopsies (n = 36)) were collected after wound debridement and cleansing with sterile physiological saline (as part of part of the routine clinical work of the hospital). Swabs were taken from open wounds by sterile cotton swabs from the base of the ulcer wound and aspirations were taken for closed lesions (abscesses and other fluctuant infected tissues) by needle aspirates after cleaning with polyvidone-iodin solution. In operated patients, intra-operative samples were obtained by infected soft tissues biopsies in the operating room. The samples were transported in sterile tubes without transport medium and were processed immediately upon arrival for bacteria recovery in the clinical laboratory as part of the routine diagnosis at the hospital.

The samples were inoculated on blood agar (Oxoid, UK, CM0271) and incubated in stove at 37 °C for 24 h. Colonies suspected to be *Staphylococcus* were subcultured on mannitol salt agar selective medium (Oxoid, UK, CM0085) for specific detection of *S. aureus*. All isolates were identified by MALDI-TOF-MS system using the standard extraction protocol recommended by Bruker (Bruker, Bremen, Germany). The identification of the *S. aureus* colonies was also confirmed by PCR of the gene *nuc* [44].

### 4.2. Antimicrobial Susceptibility Profile of S. aureus Isolates

The antimicrobial susceptibility tests on all the *S. aureus* isolates were performed by the Kirby–Bauer disk diffusion method on Mueller–Hinton agar medium (MH) (BioRad, Marne-la-Coquette, France). The antibiotic discs (BioRad, Marne-la-Coquette, France) tested were the following ones (μg/disk): penicillin (1 unit), cefoxitin (30), tobramycin (10), gentamicin (10), ciprofloxacin (5), levofloxacin (5), trimethoprim–sulfamethoxazole (1.25 + 23.75), clindamycin (2), erythromycin (15), fusidic acid (10), tetracycline (30), minocycline (30), mupirocin (200), chloramphenicol (30), linezolid (10), tigecycline (15) and rifampicin (5). The breakpoints recommended by the Clinical and Laboratory Standards Institute (CLSI) guidelines [45] were followed. The double-disc diffusion test (D-test) with erythromycin and clindamycin disks was implemented for all isolates to detect inducible clindamycin resistance. Vancomycin and teicoplanin MICs were determined using the broth microdilution method according to CLSI [45]. Isolates displaying resistance to three or more antimicrobial classes were considered multidrug-resistant (MDR).

### 4.3. Screening of Methicillin-Resistant S. aureus Isolates (MRSA)

Isolates resistant to cefoxitin (FOX, 30 µg) on MH agar according to CLSI recommendations were confirmed for the presence of the *mec*A gene by PCR technique as described previously [45,46]. *mec*A-positive isolates were considered as MRSA isolates.

### 4.4. Detection of Antimcrobial Resistannce Genes (ARGs)

The presence of genes associated with resistance to β-lactams (*mec*A and *blaZ*), erythromycin (*erm*(A), *erm*(B), *erm*(C) *and msr*(A)), tetracycline (*tet*(K)*, tet*(L) and *tet*(M)), chloramphenicol (*fexA* and *fexB*) and aminoglycosides (*aac*(*6*′)*-aph*(2″) and *ant*(4)-Ia) were analyzed using PCR and confirmed by sequencing [46].

### 4.5. Molecular Typing of Isolates

All *S. aureus* isolates were characterized by amplification and sequencing of the polymorphic region of the staphylococcal protein A-encoding gene (*spa*) [47]. The obtained sequences were analyzed using Ridom Spa-type software version 1.5.21 (Ridom GmbH, Würzburg, Germany) to determine the *spa* type. In addition, a specific PCR was carried out to identify the CC398 lineage, targeting the CC398-specific variant of *sau1-hsdS1* [48]. The clonal complex of the remaining isolates was assigned, when possible, according to the *spa*-type.

MRSA isolates were subjected to SCC*mec* typing by PCR strategy to determine the *mec* gene complex and the *ccr* gene complex as described by Zhang et al. [49]. The identification of the *agr* allele group (I–IV) was also determined by multiplex PCR as described earlier [50].

### 4.6. Occurrence of Virulence and Immune Evasion Cluster (IEC) Genes

All *S. aureus* isolates were screened using PCR for the following staphylococcal virulence genes: Panton–Valentine leukocidin (*lukS/F-PV*), toxic shock syndrome toxin (*tst*) and the exfoliative toxins (*eta* and *etb*) as previously described [51]. The immune evasion cluster (IEC) genes (*scn, chp, sak, sea* and *sep*) were examined by PCR and, based on the genes obtained, the isolates were classified into seven IEC types [51,52]. The *scn* gene (encoding the staphylococcal complement inhibitor) was used as a marker of the IEC system. Positive controls from the collection of the University of La Rioja were included in all PCR assays.

## 5. Conclusions

This is the first study to report the prevalence rate, the antimicrobial resistance profile, virulence genes and molecular typing of *S. aureus* isolates obtained from diabetic foot wounds in Tunisia. Our results indicate a high prevalence of *S. aureus* in DFIs with genetic diversity among the MSSA and MRSA isolates. A high number of MDR isolates harbored various AMR and virulence genes.

This study elucidates the recent regional epidemiological data on *S. aureus* implicated in DFIs which will be relevant for better guidelines for antibiotic use in clinical settings. These findings highlight the need for further studies focusing on the molecular surveillance of AMR for optimal management of DFI.

## Figures and Tables

**Table 1 antibiotics-11-01755-t001:** Characteristics of 64 patients included in this study.

Case Number	The Reason Why the Patient Entered the Center	Sex	Age (Years)	Country	Type of Diabetes/Duration (Years)
1	Infected plantar perforating disease of the right foot	M	62	Tunisia	II/30
2	4th left toe infection	M	53	Libya	II/5
3	Phlegmon of the sole of the right foot	M	55	Tunisia	II/12
4	Superinfection of the right transmetatarsal amputation stump	M	67	Libya	II/10–19
5	Left big toe infection	M	77	Libya	II/20
6	Infected plantar perforating disease of the right Charcot foot	M	60	Libya	II/10–19
7	Gangrene of the 4th and 5th right toe	F	68	Libya	I/20
8	Phlegmon of the left foot	M	50	Tunisia	I/28
9	Patient with sepsis (left heel infection)	M	55	Libya	II/10
10	2nd right toe infection	M	55	Libya	II/20
11	Left 4th toe stump infection	M	68	Algeria	II/20
12	Right hallux gangrene	F	68	Libya	II/30
13	Phlegmon of the plantar surface of the left foot	F	79	Libya	II/19
14	Right foot infection	M	84	Tunisia	II/10
15	Patient with sepsis (right plant gangrene)	M	71	Tunisia	II/20
16	Infected plantar perforating disease of the right foot	M	58	Algeria	II/21
17	Left foot phlegmon	F	66	Tunisia	II/27
18	Gangrene of the 2nd right toe	F	77	Algeria	II/30
19	Right big toe infection	M	67	Libya	II/+30
20	Left foot infection with purulent discharge	M	69	Lybia	II/25
21	Right hallux gangrene	M	64	Tunisia	II/25
22	Superinfection of the 4th right toe	M	80	Lybia	II/30
23	Left hallux infection	M	62	Libya	II/19
24	Infected intertrigo inter toe of the 3rd and 4th space of the right foot	F	51	Libya	II/+20
25	Left big toe infection	F	64	Tunisia	II/18
26	Phlegmon of the flexor sheaths of the left foot	M	43	Tunisia	II/14
27	Infected right foot	M	47	Libya	II/4 months
28	Infected left foot	M	43	Tunisia	II/10
29	Infected right heel	F	53	Libya	II/20
30	Charcot infection of the left foot	M	59	Tunisia	II/10
31	Right hallux gangrene	M	45	Tunisia	I/30
32	Phlegmon and plantar perforating disease of the right and left foot	M	46	Tunisia	II/15
33	Right hallux infection	M	67	Algeria	II/10
34	3rd left toe gangrene	M	56	Libya	II/25
35	3rd left toe infection	M	92	Tunisia	II/30
36	Left big toe gangrene	M	74	Tunisia	II/26
37	Infected ulceration of the 2nd right and left toe	F	69	Libya	II/15
38	Phlegmon in the sole of the left foot	M	42	Chad	II/10
39	Right big toe gangrene	M	67	Tunisia	II/25
40	Infection of the big toe and the 3rd left toe	M	63	Libya	I /50
41	2nd left toe infection	F	58	Libya	II/10–19
42	Superinfection of the amputation stump of the right hallux	F	71	Guinea	II/1
43	Left hallux infection	M	52	Libya	II/36
44	5th left toe gangrene	M	53	Libya	II/20
45	Left hallux flexor sheath phlegmon	M	74	Tunisia	II 10
46	Left big toe gangrene	M	39	Libya	II/20
47	Plantar perforating disease on Charcot foot of the left foot	M	66	Libya	II/36
48	Gangrene of the 1st and 3rd left toe	M	57	Tunisia	II/25
49	Right foot infection	M	80	Tunisia	II/15
50	Infected ulceration of the 5th right toe	M	60	Tunisia	II/36
51	Right 2nd toe gangrene	M	52	Tunisia	II/20
52	Superinfection of the left heel	M	58	Tunisia	diabetes secondary to acute pancreatitis/23
53	Left 2nd toe gangrene	F	43	Tunisia	II/10
54	Infected ulceration of the plantar surface of the right hallux	M	65	Tunisia	II/20
55	Right foot phlegmon	M	68	Tunisia	II/20
56	Left 5th toe infection	M	64	Tunisia	II/15
57	Right foot transmetatarsal amputation stump infection	M	64	Tunisia	II/40
58	Superinfection of the amputation stump of the right 1st ray	M	60	Tunisia	I/19
59	Left heel infection	F	62	Tunisia	I/30
60	Left big toe infection	F	58	Tunisia	II/20
61	Phlegmon of the 4th left toe	M	81	Tunisia	II/+20
62	Right hallux gangrene	F	66	Tunisia	II/25
63	Right heel infection	F	67	Tunisia	II/37
64	Left foot infection	M	72	Libya	II/25

F, female; M, male.

**Table 2 antibiotics-11-01755-t002:** Antibiotic resistance rate of the 15 *S. aureus* isolates from DFIs.

Antibiotic (Disc Charge)	All Isolates (Total = 15) n (%)	MRSA (Total = 6) n (%)	MSSA (Total = 9) n (%)
penicillin	14 (93.3%)	6 (100%)	8 (88.9%)
cefoxitin	6 (40%)	6 (100%)	0
tobramycin	3 (20%)	3 (50%)	0
gentamicin	3 (20%)	3 (50%)	0
ciprofloxacin	2 (13.3%)	2 (33.3%)	0
levofloxacin	2 (13.3%)	2 (33.3%)	0
trimethoprim–sulfamethoxazole	2 (13.3%)	2 (33.3%)	0
clindamycin	1 (6.7%)	1 (16.7%)	0
erythromycin	2 (13.3%)	0	2 (22.2%)
fusidic acid	11 (73.3%)	6 (100%)	5 (55.6%)
tetracycline	4 (26.7%)	4 (66.7%)	0
minocycline	1 (6.7%)	1 (16.7%)	0
mupirocin	1 (6.7%)	1 (16.7%)	0
chloramphenicol	1 (6.7%)	1 (16.7%)	0
linezolid	0	0	0
tigecycline	0	0	0
vancomycin	0	0	0
teicoplanin	0	0	0
rifampicin	1 (6.7%)	1 (16.7%)	0

**Table 3 antibiotics-11-01755-t003:** Characteristics of the six MRSA and nine MSSA isolates recovered from diabetic foot infections in this study.

Strain	Sample Type	Molecular Typing				Antimicrobial Resistance		Virulence Genes	IEC ^f^
		*Spa* Type	CC ^c^	Agr-Type	SCC*mec*-Type	Phenotype ^e^	Genotype		
**X3653 ^a^**	Aspiration	t311	CC5	IV	V	PEN, FOX, CIP, LVX, FA	*mecA, blaZ*		E
**X3655**	Aspiration	t037	CC8	IV	IV ^b^	PEN, FOX, SXT, FA, TET, CHL	*mecA, blaZ, tet(M), fexA*		G
**X3656**	Deep swab	t127	CC1	III	ND ^d^	PEN, FOX, TOB, GEN, SXT, FA, TET, MUP	*mecA, blaZ, tet(L), tet(K), aac6′-aph2”*	*lukS/F-PV*	D
**X3657**	Deep swab	t15077		III	ND	PEN, FOX, FA	*mecA*		E
**X3659**	Deep swab	t688	CC5	III	III	PEN, FOX, TOB, GEN, CIP, LVX, FA, TET, MIN, RIF	*mecA, blaZ, tet(M), tet(K), aac6′-aph2”*		D
**X3654**	Deep swab	t084	CC15	II	V	PEN, FOX, TOB, GEN, FA, TET	*mecA, blaZ, tet(K), aac6′-aph2”, ant4′-la*		C
**X3694**	Deep swab	t127	CC1	III	-	PEN, FA	*blaZ*		D
**X3695**	Deep swab	NC ^b^		III	-	PEN	*blaZ*		-
**X3697**	Tissue biopsy	t118		III	-	PEN, ERY, FA,	*blaZ, erm(B), msr(A)*	*eta*	-
**X3698**	Tissue biopsy	t355		III	-	PEN	*blaZ*	*lukS/F-PV*	E
**X3693**	Tissue biopsy	t091		III	-	PEN	*blaZ*		G
**X3699**	Deep swab	t012	CC12	III	-	PEN, ERY, CLI *	*blaZ, erm(A), erm(B), msr(A)*	*tst*	A
**X3700**	Tissue biopsy	t223	CC22	III	-	PEN, FA	*blaZ*		B
**X3692**	Tissue biopsy	t571	CC39	III	-	FA	-		C
**X3696 ^a^**	Deep swab	t127	CC1	I	-	PEN, FA	-		D

^a^, isolates obtained from the same patient; ^b^ NC, novel combination: the repetitions detected in *spa*-type were as follows: r03-r16-r21-r17-r23-r12; ^c^ CC: they were assumed according to the *spa* type, except for CC398 that was determined by specific PCR; ^d^ ND, non-determined; ^e^ PEN, penicillin; FOX, cefoxitin; ERY, erythromycin; TET, tetracycline; CIP, ciprofloxacin; LVX, levofloxacin; FA, fusidic acid; SXT, trimethoprim–sulfamethoxazole; TOB, tobramycin; GEN, gentamycin; MIN, minocycline; CHL, chloramphenicol; MUP, mupirocin; RIF, rifampicin; CLI *, inducible resistance; ^f^ IEC, immune evasion cluster.

## Data Availability

Not applicable.

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
