# Peer review of "Methicillin-Resistant Staphylococcus aureus from Diabetic Foot Infections in a Tunisian Hospital with the First Detection of MSSA CC398-t571"

_antibiotics, 2022, doi:10.3390/antibiotics11121755_

Round 1

Reviewer 1 Report

Abstract:

·         Line 17: Please use full name as “Kirby-Bauer disk diffusion susceptibility test”.

·         Line 21: The isolates heberged different resistance genes:……..? Please amend this.

·         Please mention the conclusion of this study in abstract also.

Introduction:

·         The authors need to revise introduction carefully to make it more fruitful.

·         Please replace “beta-lactam” with “β-lactam” in whole manuscript.

·         In introduction, line 44-49, blaZ gene is also contributing to resistance in bacteria against β-lactam antibiotics. Please amend your introduction and explain the role β-lactamases in AMR also.

·         Line 50-53, Strong statements have been given in 3rd paragraph of the introduction without references. Please provide references

·         Report the epidemiology of DFI caused by S. aureus in Tunisia and in neighbouring countries. Please show the study gaps.

·         Check reference number 13: Use an appropriate reference that support the line 54-55.

·         In last paragraph of introduction, Objective of this study is not clear, please revise this.

Results:

·         Line 63: how many males and female have type 1 or type-2 diabetes? Please revise this separately for both males and female.

·         During September 2019 to October 2020, just 64 patients having DFI among which only 15 were positive to S. aureus. Among positive samples, MRSA were detected only in six sample. Number of samples are not sufficient to conclude anything that will be beneficent to physicians, researchers or clinical setting.

·         Line 66-72: “In this study, S. aureus 21.9% (n=14/64) was detected in patients suffering from DFI”. Pease revise this portion accordingly.

·         Line 75: What is meant by all isolate? Please give the number as: A total of 64 isolates showed resistance to at least one antibiotic”. Please amend this.

·         Line 80-81: What about teicoplanin? Please check this.

·         Please mention the data of AMR and genetic prevalence among MRSA and MSSA, separately.

·         Line 100: PVL??? Please use the full name with abbreviation and then you can use only abbreviation.

Methodology

·         Line 177: How did you collect 83 samples from 64 patients? please revise this

·         How did you characterize the DFI and diabetes either type 1 or 2?

·         Please give the description of the Hospital? What is the bed capacity? Please refer the SROBE checklist to ensure each section of the manuscript is well written (https://www.strobestatement.org/checklists/)

·         Line 183: operating theatre?

·         Not clear regarding sample collection, transportation and preservation: How did you collect the samples by sterile swab either via aspiration or biopsy? How did you transport the samples into microbiology lab? Did you preserve the sample? If yes, then for how many days and what was the temperature?

·         Line 186: Did you incubate the culture plates at room temperature or in incubator? Please clarify this.

·         Line 186-187: How did you suspect colonies to be Staphylococci spp.? Did you perform gram-staining or catalase test for Staphylococci?

·         Line 188-190: What was the conditions applied for nuc gene PCR? Please give details.

·         Also provide gel pictures of PCR.

·         Line 192: In the method and result part, the authors used disk diffusion to do the antimicrobial activity. Is it about the measurement of zone inhibition or something else? This needs to be clarified.

·         Provide the manufacturer details of each material used (Company, City, Country). Line 193: Mueller-Hinton agar medium?

·         Line 203-204: Please mention the phenotypic and genotypic identification of MRSA as ………….?

·          Line 218: please give the details of multiples PCR. What concentrations of dNTPs, Taq polymerase, Minerals and other used?

Discussion:

·         Revise the discussion accordingly. What did you expect?

·         Avoid much repetitions of the results, please follow the STROBE guidelines

·         Line 126: In our study, the prevalence of S. aureus detected was 21.9%. This rate was higher than those detected in the two earlier studies conducted in Tunisia (9% and 17%). Please revise this.

·         Line 126-129: Your findings are in contrast of other authors studies. Why? Please give statement.

·         Line 136-139: there is no need to repeat the results. Please avoid too much repetition of results in discussion.

·         Line 162; Panton valentine leucocidin toxin (PVL)???

Author Response

I appreciate the comments of reviewers that help to improve the quality of the manuscript. thanking you for identifying the weaknesses in our paper and providing us the opportunity to strengthen our research prior to publication

I enclose the answer to all comments of reviewers and changes in manuscript have been marked in yellow letters. I hope that this new version will be more suitable for publication in antibiotics

Thanks for all your interesting comments.

you found the responses in document in attachement

Reviewer 2 Report

The manuscript titled  “Methicillin-resistant Staphyloccocus aureus infection from diabetic foot ulcers in a Tunisian hospital, with first detection of  MSSA CC398-t571” lays out a very interesting analysis of antimicrobial susceptibility phenotypes/genotypes and virulence of S. aureus isolated from diabetic foot ulcer patients in Tunisia hospital. The work is well performed and easy to follow. I recommend for acceptance with minor comments addressed below:

1.- In results: “Since the diabetic foot center is international, the patients were from different African countries (Tunisia, Algeria, Libya, Chad and Guinea)”. I suggest some short note about percentage of infection for patients of each country to complete the results at lines 70-71.  

Author Response

I appreciate the comments of reviewers that help to improve the quality of the manuscript. thanking you for identifying the weaknesses in our paper and providing us the opportunity to strengthen our research prior to publication

I enclose the answer to all comments  and changes in manuscript have been marked in yellow letters. I hope that this new version will be more suitable for publication in antibiotics

The manuscript titled  “Methicillin-resistant Staphyloccocus aureus infection from diabetic foot ulcers in a Tunisian hospital, with first detection of  MSSA CC398-t571” lays out a very interesting analysis of antimicrobial susceptibility phenotypes/genotypes and virulence of S. aureus isolated from diabetic foot ulcer patients in Tunisia hospital. The work is well performed and easy to follow. I recommend for acceptance with minor comments addressed below:

1.- In results: “Since the diabetic foot center is international, the patients were from different African countries (Tunisia, Algeria, Libya, Chad and Guinea)”. I suggest some short note about percentage of infection for patients of each country to complete the results at lines 70-71.  

Answer: Thanks a lot for your comments, suggestion has been followed.

Round 2

Reviewer 1 Report

The authors have addressed my comments/corrections.